# Frictional Properties of the TiNbTaZrO Orthodontic Wire—A Laboratory Comparison to Popular Archwires

**DOI:** 10.3390/ma14216233

**Published:** 2021-10-20

**Authors:** Krzysztof Schmeidl, Michal Wieczorowski, Katarzyna Grocholewicz, Michal Mendak, Joanna Janiszewska-Olszowska

**Affiliations:** 1Department of Interdisciplinary Dentistry, Pomeranian Medical University in Szczecin, Powstancow Wielkopolskich 72, 70-111 Szczecin, Poland; krzysztof.schmeidl@gmail.com (K.S.); katgro@pum.edu.pl (K.G.); jjo@pum.edu.pl (J.J.-O.); 2Division of Metrology and Measurement Systems, Poznan University of Technology, Piotrowo Street 3, 60-965 Poznan, Poland; michal.mendak@put.poznan.pl

**Keywords:** Gummetal, dynamic friction, TiNb, orthodontic archwire

## Abstract

Background. This study aimed to determine the kinetic frictional force (FF) of the recently produced TiNbTaZrO (Gummetal) orthodontic wire and compare it to the widely used wires of stainless steel (SS), nickel-titanium (NiTi), cobalt-chromium (CoCr) and titanium-molybdenum (TiMo) alloys. Methods. Five types of 0.016″ × 0.022″ wires were ligated with elastic ligatures to 0.018″ × 0.025″ SS brackets. The dynamic FFs between the brackets and ligated wires were measured utilizing a specialized tensile tester machine. Prior sample sizes for different archwires were conducted using power analysis for the general linear models. The existence of significant differences in FF between examined materials was initially confirmed by the one-way analysis of variance (ANOVA) with further evidence of pairwise differences by Tukey’s Honest Significant Difference test. Results. The pairwise differences between means of kinetic FFs for NiTi, CoCr, and Gummetal wires were not statistically significant (adjusted *p*-value > 0.05). Stainless steel alloy presented the lowest FF values significantly different from other groups (adjusted *p*-value < 0.05). On the contrary, TiMo wires showed significantly greater FFs (adjusted *p*-value < 0.05) than other alloys. Conclusions. Gummetal orthodontic wire exhibits similar frictional resistance as NiTi and CoCr wires. Bendable TiNbTaZrO wire might be used for sliding mechanics due to its favorable frictional properties.

## 1. Introduction

Friction is the force that opposes motion between any surfaces that are in contact [1]. Frictional force (FF) in orthodontics occurs at multiple contact points along the bracket-archwire interface during multibracket treatment. Brackets bonded on different tooth surfaces transfer forces to teeth [2,3]. Orthodontic tooth movement depends on static and dynamic friction [4]. Sliding between wire and bracket in the oral cavity occurs at a low velocity. Variables affecting friction between the components of multibracket appliances are related to the type of archwire, the bracket, the ligation system, and biological factors [5,6].

Different alloys are used for producing orthodontic wires required in the active phase of orthodontic treatment. Clinicians should be aware of frictional resistance of archwires, especially during *en masse* movement or canine retraction. When sliding mechanics are utilized, FF is generated between the wire, bracket, and ligature, impeding tooth movement during the phase of retraction and transmitting forces to the posterior teeth. These forces negatively affect the anchorage requirement, potentially resulting in loss of anchorage [7]. The ideal orthodontic wire should provide a low coefficient of friction to move the teeth efficiently and freely.

Stainless steel (SS) wires are widely known for their high stiffness and low friction properties compared to other archwire materials [8]. The use of a combination of SS archwires and brackets has been the gold standard for orthodontists to utilize during sliding mechanics. Stainless steel archwires are also considered the reference material for assessing the mechanical properties of newly fabricated archwires [9,10].

Cobalt-chromium (CoCr) wire emerged in the 1960s [11]. One advantage of using it for orthodontic applications is its low hardness. After being shaped, the CoCr archwire can be hardened by heat treatment, which substantially increases its strength. For years, SS and CoCr alloys were the standards in orthodontic treatment.

Nitinol was developed in the early 1960s and introduced to orthodontics in the late 1970s [12]. Orthodontic nickel-titanium (NiTi) wires are widely used. They have simplified the initial phase of orthodontic treatment due to their low forces over a wide range of activation and superelastic properties [13]. However, the low deformability of NiTi archwires limits their use in the final phases of orthodontic therapy.

Titanium-molybdenum (TiMo) is a beta-titanium (β-Ti) alloy introduced to orthodontics in the 1980s. It combines high strength and springiness, making it a good choice for finishing (especially rectangular wires used in the late stages of multibracket orthodontic treatment). Beta-titanium lacks the stiffness and formability of CoCr; however, it is not suitable for applications that require the flexibility of an archwire. Another drawback of TiMo wires is their high surface roughness, which produces high friction between the bracket and archwire [14]. The development of new β-Ti wires and other titanium alloys has rapidly increased recently, partly because of the high biocompatibility of these nickel and chromium-free archwires [15].

In previous studies of frictional resistance of orthodontic wires, TiMo wires generated the highest friction values, followed by NiTi and SS wire, when using the SS brackets [16]. Another study stated that SS wires performed the least amount of friction, followed by CoCr, NiTi, and β-Ti wires [17].

Gummetal is a recent multifunctional β-Ti alloy. It consists of titanium, niobium, tantalum, zirconium, and oxygen (TiNbTaZrO), making it bioinert [18,19,20]. The abovementioned metals are placed in groups IVa and Va of the periodic table of elements. Gummetal was developed in 2001 at the Metallurgy Research Section of Toyota Central R&D, Inc. in Japan [21]. The chemical composition of Gummetal (Ti-23Nb-0.7Ta-2Zr-1.2O) was based on atomic valence theory. To achieve its rare characteristics, the alloy is intensively cold-worked. Special characteristics of Gummetal arise from the juxtaposition of electronic magic numbers: Bo value of 2.87, compositional average valence electron number of 4.24, and Md value of 2.45 eV [22]. According to Hasegawa, Gummetal wire can reduce friction between the archwire and metal brackets by up to 50% compared with other titanium wires [22]. It exhibits a very low Young’s Modulus constant with the temperature and high tensile strength (which is extremely rare) and provides lower force than NiTi and β-Ti archwires. Manufacturers state that the dislocation motion of the crystal (plastic deformation) is controlled completely, which makes Gummetal unique [23].

The authors of this research have found only a single article comparing the FFs of Gummetal, NiTi, and TiMo orthodontic wires [24]. Besides, in their recently published review, the present study’s authors have not found any published studies comparing FFs of Gummetal with other conventional wires [25]. Thus, the objective of this study was to determine the kinetic FFs of Gummetal wire and compare them with the abovementioned well-known archwire alloys.

Although according to the theory, the FF is not directly related to the surface, the authors decided to analyze the topography of all five orthodontic wires. Materials were intentionally placed in contact, and the nature of the contact is linked directly to the layout of surface asperities, which is the subject of functional surface analyses in many areas of science [26,27].

## 2. Materials and Methods

In the present study, 50 SS brackets (Discovery, Dentaurum) with declared slot sizes of 0.018 × 0.025″ (0.018″ slot) and Roth prescription for the maxillary right canine were used. Five types of 0.016 × 0.022″ 10 cm long straight wires (*n* = 50), including (ten specimens each): SS (Remanium, Dentaurum, Ispringen, Germany), CoCr (Elgiloy Blue, RMO, Denver, CO, USA), NiTi (Nickel Titanium, G&H, Franklin, IN, USA), TiMo (BetaForce, Ortho Technology, West Columbia, SC, USA), and TiNbTaZrO wires (Gummetal, JM Ortho Corporation, Tokyo, Japan) were ligated to the brackets with elastic modules (Alastik Easy-To-Tie, 3M, Saint Paul, MN, USA).

Each bracket was bonded on a steel plate S235JR 1,4 mm thick, 50 mm wide, and 115 mm long, using cyanoacrylate adhesive (Kemsin KC-1200, Chemkon). Bracket prescription characteristics were minimized by supporting the bracket with a 0.018 × 0.025″ SS straight wire jig (Remanium, Dentaurum). The jig was removed once the adhesive had hardened. All the bonding procedures, for standardization purposes, were carried out by the same operator. The measurement condition was taken at a dry state at room temperature. Then, each wire was ligated to the bracket with elastic ligature so that the lower, shorter end of the wire was left free.

Following the bonding and ligating procedures, each model was mounted in the base handle of the MultiTest 2.5-i testing machine (Mecmesin Ltd., Horsham, UK) (Figure 1). The steel plate was placed in the handle of the machine and then fixed with straight wire perpendicular to the machine’s base and parallel to the bracket’s slot.

Afterward, 10 mm of the upper end of the wire was fixed to the special tension-loading cell of the tensile measuring machine with a range of up to 10 N loadcell capacity and full-scale accuracy of ±0.1%. Subsequently, the wire was pulled through at a crosshead speed of 10 mm/min for a distance of 20 mm. The machine was recording 1000 measurements of FF per second. The kinetic FF was measured by averaging the FFs exerted during the 120 s long-lasting wire movement. Recorded measurements of dynamic FF for each of 50 wire-ligature-bracket combinations were then saved on a PC employing Emperor (Mecmesin Ltd.) force testing software.

The statistical analysis was undertaken using IDE version 1.4.1106 (RStudio) and R version 4.0.4 (R Core Team) software. For the prior sample size estimation, the authors used power calculations for the general linear models. Assuming the degrees of freedom for the numerator u = 4 (for 5 groups), effect size f2 = 0.4 (we used a medium value for a priory testing), a significance level of 5%, and large power value = 0.75, the total sample size for the present experiment should have equaled at least 48, thus having 10 samples per group met the conditions [28,29].

To carry out a statistical analysis of the data, the authors have proceeded from two classical hypotheses:A null hypothesis: there is no difference in kinetic FF between groups.The alternative hypothesis proposes that there is a difference at least between one pair of groups.

Several statistical tests were performed to test the hypotheses. However, some tests were designed to work with small or medium size samples (up to 5000). Therefore, when performing statistical tests, the mean values of the sample groups were used instead of the whole groups of samples. The use of sample means was justified because each data distribution group, as seen in the next section, generally corresponded to the normal one.

A one-way analysis of variance (ANOVA) has been performed to determine if there were differences between the groups. ANOVA is an extension of independent two-samples *t*-test for comparing means in a situation where there are more than two groups.

However, the ANOVA test is based on the assumptions of the homogeneous variance across groups and the normally distributed data. Levene’s test for group homogeneity estimation has been used to check the first assumption. This test is less sensitive to departures from normal distribution compared to Bartlett’s test [29]. For checking the second (normality) assumption, the Shapiro–Wilk’s test on the ANOVA residuals has been performed.

A general linear hypothesis test was applied to specify between which group of samples, potentially, there were statistical differences in kinetic FFs.

In order to demonstrate the differences visually, the post hoc Tukey’s Honest Significant Difference (HSD) test was performed. That test relies on a set of confidence intervals on the differences between the means of the levels of a factor with the specified family-wise probability of coverage. The intervals were based on the studentized range statistic, Tukey’s HSD method.

There are many methods for measuring surface topography, divided into two main groups: contact and contactless. Since it is difficult to achieve consistency between methods using different physical principles [30], it was decided to use a focus variation microscope with an appropriately selected lighting system [31], as shown in Figure 2.

Focus variation microscopy sharpens the surface image to measure the surface unevenness. The measured object is illuminated by light with appropriate modulation, transmitted by optics, and focused on the surface. The reflected light returns through the optical system and reaches the digital detector searching for the focus beam. The surface image is shaped by an optical system capable of obtaining both photometric (brightness, color, etc.) and geometric (distances, shape) information [32]. For the surface topography reconstruction, only those places where the data from the detector coincide with the data from the focus area will be stored.

In the present study, the measurements were made for representative samples of each material utilizing Alicona IF G5 focus variation microscope. After obtaining raw measurement data, a procedure was carried out to fill in non-measured points, which could be problematic with many optical measurements [33]. Subsequently, the shape was then removed by a third-degree polynomial and filtration of the surface waviness employing the Gaussian filter. A cut-off (compliant with a nesting index) of 0.25 mm was used. For measurements, 50× magnification was chosen with coaxial illumination (no polarization), 20 nm vertical resolution, and 2.15 μm horizontal resolution. Sampling distance was equal to 0.176 μm with a total measurement area of 2.629 mm × 0.611 mm.

## 3. Results

The distribution of the kinetic FFs by the samples compared has been presented in Table 1.

That being the case, the Remanium’s and Gummetal’s interquartile ranges (IQRs) are comparatively narrow. This suggests that overall data have a high level of agreement with one another. On the contrary, Elgiloy data is the most dispersed. Additionally, Remanium and BetaForce medians do not intersect with other group’s IQRs, then there are differences between these groups. Medians for Nickel-Titanium, Elgiloy Blue, and Gummetal lie within one another’s IQR boundaries.

Figure 3 presents the FF distribution data for each group as a histogram. A normal distribution curve was superimposed with mean and standard deviations (SDs) from Table 1 to visualize and assess how the histograms deviated from a normal distribution.

All groups have bell-shaped (unimodal) histograms, except Nickel-Titanium, which has a bimodal one. Positively skewed histograms describe Remanium and Nickel-Titanium with highly and moderately kurtosis, respectively. Also, kurtosis but a negative one is present on Elgiloy Blue histogram. The values of asymmetry and excess for BetaForce and Gummetal groups of samples are insignificant; moreover, these groups have the lowest SDs.

Initially, based on the analysis of descriptive statistics, the values for BetaForce, Gummetal, and Remanium respectively most closely satisfy the criteria for normal distributions.

In general, there are some deviations from the normal distribution. However, due to the very large sample size (*n* = 120,000) and 10 independent samples in each group, the influence of outliers’ presence remains negligible. The mean values of the sample groups used to perform statistical tests are presented in Table 2.

Summarizing the analysis of variance group sample means is given in Table 3.

Because the *p*-value is less than the significance level of 0.05, it indicates the differences between the groups.

The results of Levene’s test have been presented in Table 4.

The test reveals a *p*-value greater than 0.05, indicating no significant difference between the group variances. Therefore, the homogeneity of variances in the different groups has been confirmed.

The result of the Shapiro–Wilk’s test output obtained W = 0.98355, *p*-value = 0.7078, so far as *p*-value is greater than 0.05, hence, no indication that normality is violated.

Based on the results of a one-way ANOVA test along with Levene’s and Shapiro–Wilk’s tests, it can be assumed that there is a difference in kinetic FFs between some groups. To specify between which ones, a general linear hypothesis test was applied (Table 5).

According to the data in Table 5, a statistical difference is significant for groups with Pr(>|t|) less than the *p*-value (0.05).

Confidence intervals estimated by performing the post hoc Tukey’s HSD test are presented in Figure 4.

Data presented in Figure 4 report that three out of ten pairwise group differences (where confidence interval crosses the dotted line containing zero) are considered non-significant.

Based on the information regarding Pr(>|t|) from Table 5, lower and upper confidence interval values from Figure 3, results of the statistical significance of the differences between the sample group means are presented in Table 6.

Seven out of ten pairwise group comparisons turned out to be statistically significant. Remanium and BetaForce group sample means both showed a significant difference between each other and between all the other groups. On the other hand, no statistically significant differences were found between pairwise comparisons of Nickel-Titanium, Gummetal, and Elgiloy Blue sample group means. Among the studied groups, the lowest values of kinetic FFs were exerted by Remanium, the highest ones—by BetaForce; the medium forces were shown by Elgiloy Blue, Nickel-Titanium, and Gummetal.

A topographical analysis of wires from the compared materials was also carried out. The images of each of them, together with their topographical maps, are shown in Figure 5.

Results of topography and roughness parameters are presented in Table 7.

All the parameters were evaluated according to ISO 4287 [34] and ISO 25178 [35]. The following parameters were presented:RSm—mean width of the roughness profile elementsSq—root mean square height of the scale-limited surfaceSsk—skewness of the scale-limited surfaceSz—maximum height of the scale-limited surfaceSk—core heightSpk—reduced peak heightSvk—reduced dale heightSmr1—material ratio at the intersection line separating hills from the core surfaceSmr2—material ratio at the intersection line separating dales from the core surface

## 4. Discussion

Friction in orthodontic therapy is a considerable clinical challenge, which must be well controlled and understood considering that it cannot be eliminated from the multibracket treatment. Using materials providing lower levels of friction reduces the force lost during the sliding of orthodontic wire. Thus, forces transmitted to the periodontal ligament are decreased, and overloading might be avoided [36].

The frictional force is a part of the resistance to sliding (RS), such as when a bracket moves along an archwire during orthodontic therapy. Kusy and Whitley [37] divided RS into three components: FFs, due to contact of the bracket surfaces and wire; binding, which is created when the archwire flexes or the tooth tips and contacts between the wire and the bracket corners occurs; and notching, when plastic deformation of the wire cross-section happens at the bracket-wire corner interface, which often takes place in clinical situations. In the study by Nishio et al. [38] the values of FF were directly proportional to the angulation increase between the wire and the bracket. Orthodontists should be well aware of all the factors influencing efficient tooth movement.

In a single study investigating frictional properties of Gummetal found, Takada et al. [24] reported that TiMo exhibited significantly higher frictional values than Gummetal and that NiTi archwires showed comparable frictional characteristics to TiNbTaZrO alloy. Similar results have been reported in the present study.

Alfonso et al. [39] reported that the frictional resistance is affected by the hardness of archwire alloys and published results that present a linear relationship between the hardness of different archwire alloys and the friction coefficients. These findings indicate that the frictional resistance of Gummetal wire might be slightly greater because of its lesser hardness and lower elastic modulus compared to NiTi archwire [24].

In the current work, the authors decided to use conventional elastomeric ligatures as the ligation method. As in previous similar studies, elastic ligatures were placed immediately before the tests, with no decay caused by exposure to time or a humid environment [40]. This approach provides impartial and comparable conditions for testing FFs in vitro [41,42].

In the present study, care was taken to place each bracket in a position so that the bracket slot was passive with respect to 0.016 × 0.022″ straight wires mounted on a steel plate. However, the authors are aware that minor non-linearity of the wire and malalignment of the orthodontic bracket could not be fully controlled and might have affected obtained results.

It is known that bracket sets, series, and single brackets are characterized by imperfections [43] that might affect the bracket slot’s structure, dimensions, roughness, and shape, e.g., divergence, convergence, parallelism, and rounding off their inner walls [44,45]. If contact between bracket and archwire is looser and smoother, friction is reduced; on the other hand, tight contact and rougher surface result in increased friction values [42]. Moreover, differences exist between declared and actual orthodontic bracket slot sizes [43,44]. Thus, fifty brackets from one producer were used in this experiment to improve the statistical control by decreasing the impact of confounding variables.

Each of the 50 tests performed in the present study comprised a new section of a wire and a new bracket to avoid mechanical wear signs of the brackets or the wire. Drawing a wire through the bracket results in a plastic displacement of the surface and the near-surface material and the detachment of particles responsible for wear debris [46]. Thus, using one bracket for many tests might have influenced obtained FFs results.

Forces needed to overcome friction in vitro are higher than those in vivo, as claimed by Ho and West [47]. Lubrication plays a role in reducing FFs between brackets and wires during in vitro tests with different types of human and artificial saliva [48]. This fact might be a possible limitation of the present in vitro study performed in dry conditions. Fortunately, in vitro studies of archwires performed in dry conditions present rankings of FFs and order of frictional values similar to wet conditions and can provide orthodontists valuable and relevant clinical information [37]. On the other hand, it is difficult to be completely certain how precisely laboratory equipment could recreate the same in vivo situation with the response of periodontal ligament to orthodontic forces [49]. At the moment, tooth movement cannot be completely imitated in vitro [36,50].

The topographic analysis of the surfaces showed that the Gummetal archwire did not deviate from the others in fundamental respects. All wires after processing presented a periodical structure, which was depicted by the values of RSm parameters. Their variability exhibited a change in machining parameters while maintaining the process. The greatest asperities occurred for Elgiloy Blue, as shown by the amplitude values of the mean (Sq) and maximum (Sz) parameters. These values were several times higher than those obtained for BetaForce, which in turn were the smallest. The difference with Gummetal was that a positive value of skewness (Ssk) was obtained, with negative values for other materials. That demonstrated the slight domination of individual, high peaks over the valleys. However, these differences were not substantial, and one has to bear in mind the stochastic component. The last group of analyzed parameters was obtained from the material ratio curve. These parameters [51] are commonly used for assessing contact surfaces subject to the wear process. Here, also for Gummetal wire, the authors found confirmation of the domination of peaks over valleys, as the Spk was larger than Svk. For other materials, this relationship was reversed. The surfaces presented generally poor properties for maintaining the lubricating medium, which can be observed from the low values of Svk in relation to Sk. The relatively high value of Smr2 also shows the low impact of the valleys for Gummetal. However, all the parameters presented are similar in nature for all the materials analyzed, making future wear tests independent of topography.

## 5. Conclusions

Gummetal wire unveiled similar frictional resistance as the CoCr and NiTi archwires. The frictional properties presented by Gummetal alloy were superior to the TiMo but inferior to the SS alloy wire. There are no significant differences in surface topography for different materials, which will make the wear test much more reliable. More in vitro and in vivo studies are necessary to find the best use for Gummetal archwire in orthodontic treatment.

## Figures and Tables

**Figure 1 materials-14-06233-f001:**
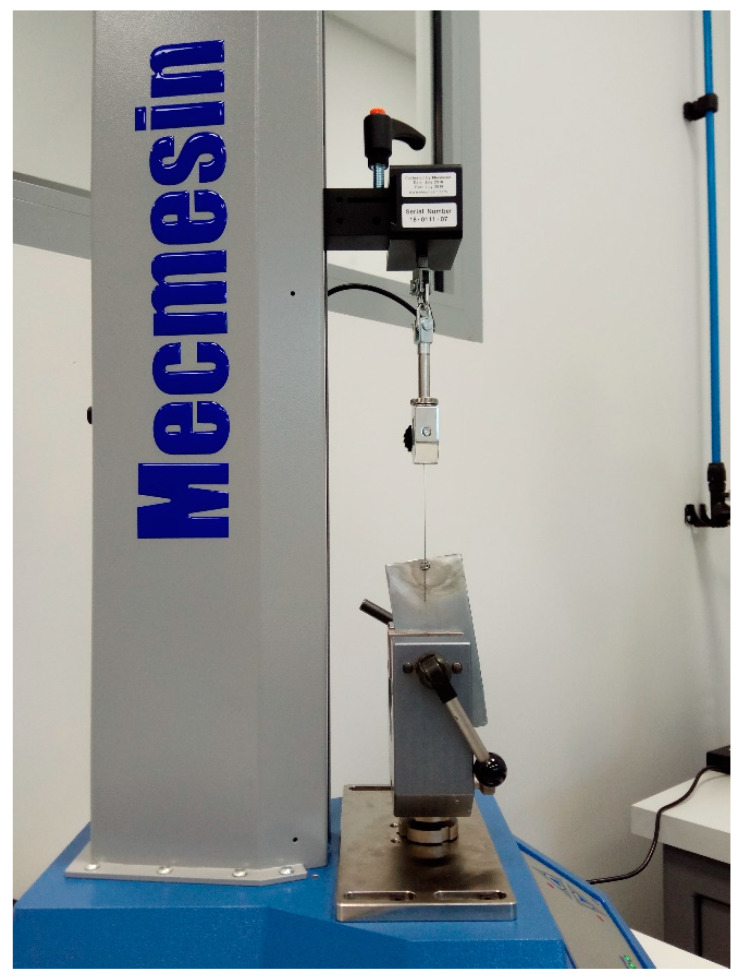
A model mounted in the base handle of the force testing machine.

**Figure 2 materials-14-06233-f002:**
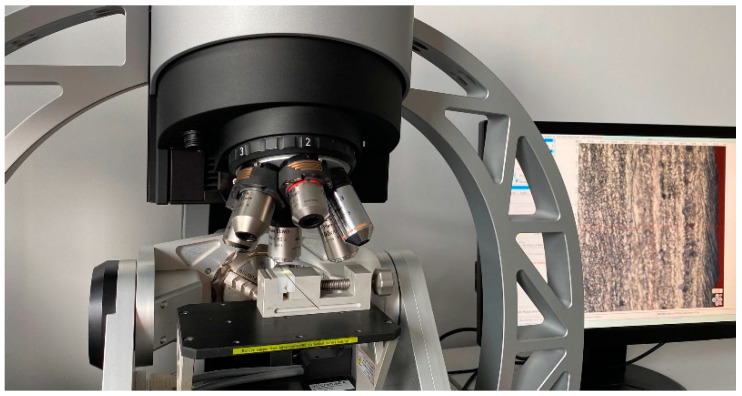
A focus variation microscope used for topography measurements with a wire in a holder.

**Figure 3 materials-14-06233-f003:**
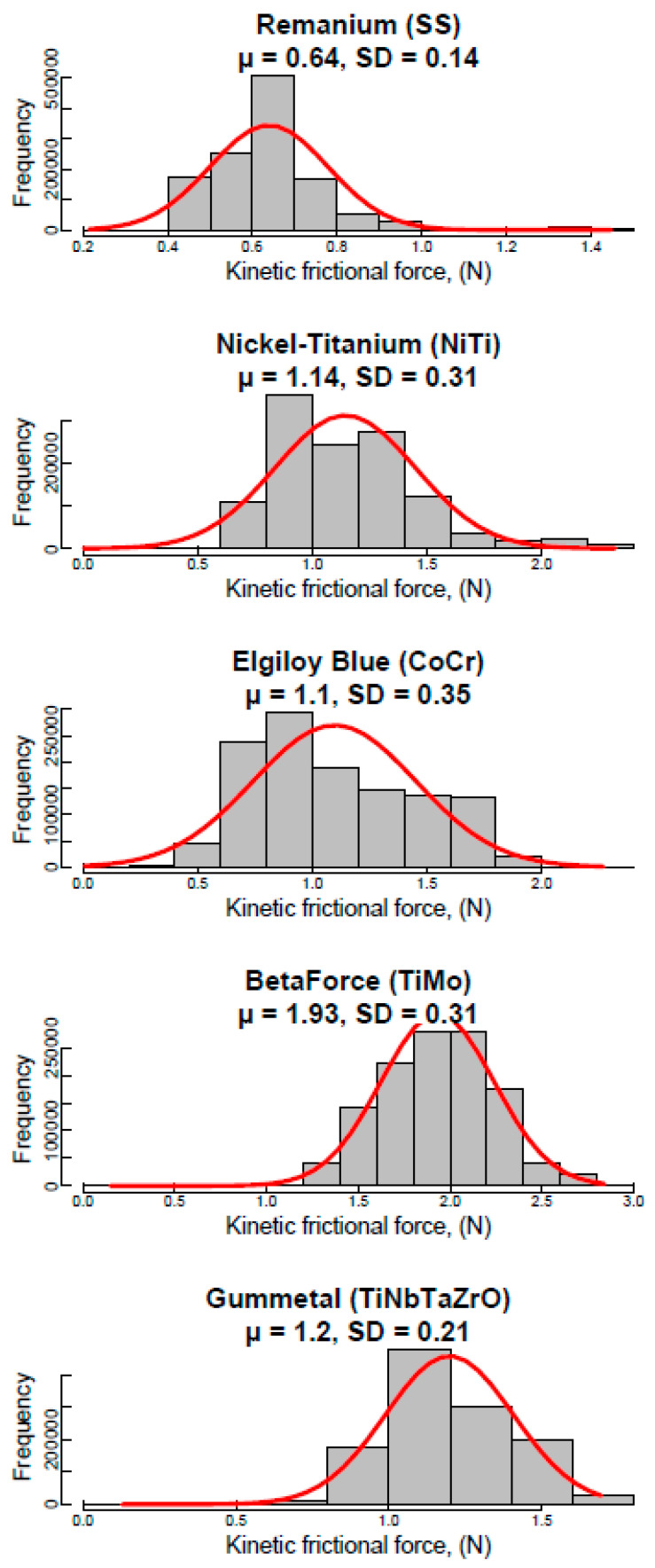
Histogram with normal curve per group of samples (*n* = 1,200,000) presenting the empirical distribution of kinetic FF (unit; N).

**Figure 4 materials-14-06233-f004:**
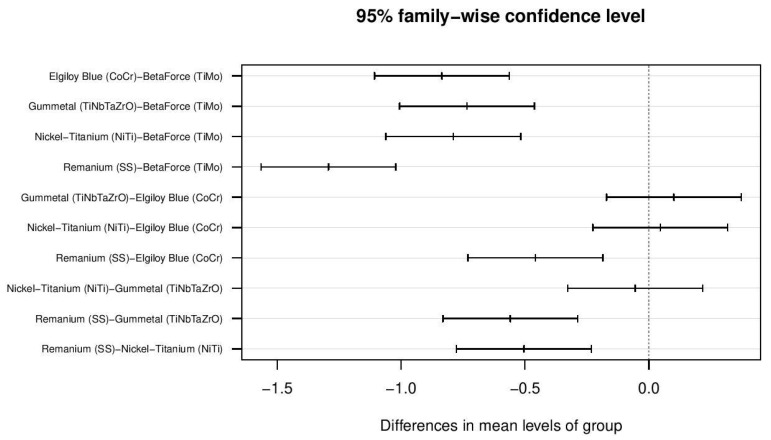
Pairwise group means differences in the form of confidence intervals.

**Figure 5 materials-14-06233-f005:**
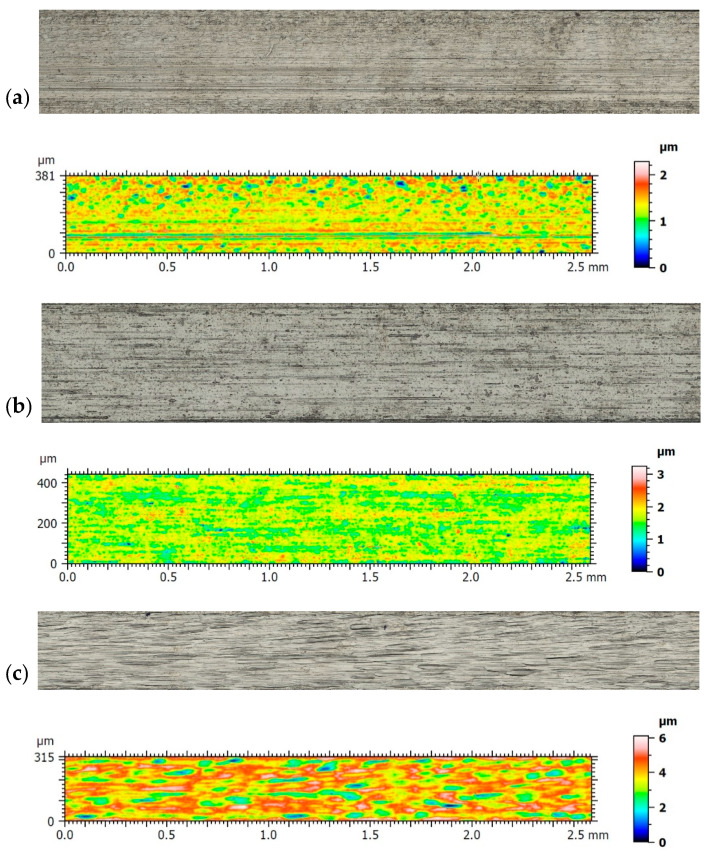
Images of different wires: (**a**) Remanium (SS); (**b**) Nickel-Titanium (NiTi); (**c**) Elgiloy Blue (CoCr); (**d**) BetaForce (TiMo); (**e**) Gummetal (TiNbTaZrO).

**Table 1 materials-14-06233-t001:** Distribution of the kinetic FF by the groups of samples (unit; N).

Group ^1^	Mean	SD	Median	TM ^2^	Min	Max	Skew	Kurtosis	SE	IQR	Q1	Q3
Remanium (SS)	0.639	0.139	0.640	0.628	0.216	1.446	1.693	6.502	<0.0001	0.141	0.548	0.689
Nickel-Titanium (NiTi)	1.143	0.306	1.087	1.114	0.005	2.319	1.020	1.508	<0.0001	0.399	0.917	1.316
Elgiloy Blue (CoCr)	1.097	0.354	1.014	1.078	0.011	2.267	0.409	−0.932	<0.0001	0.569	0.813	1.382
BetaForce (TiMo)	1.932	0.307	1.930	1.934	0.152	2.839	−0.138	0.253	<0.0001	0.431	1.727	2.158
Gummetal (TiNbTaZrO)	1.198	0.208	1.181	1.192	0.129	1.694	0.192	0.060	<0.0001	0.244	1.064	1.308

^1^—Sample size per group *n* = 1,200,000. ^2^—Trimmed mean.

**Table 2 materials-14-06233-t002:** Descriptive statistics of kinetic FF by group sample means (unit; N).

Group	Sample									
	1	2	3	4	5	6	7	8	9	10
Remanium (SS)	0.45	0.88	0.5	0.61	0.64	0.72	0.55	0.73	0.65	0.65
Nickel-Titanium (NiTi)	1.35	1.04	0.94	0.93	1.31	1.66	0.84	0.91	1.06	0.86
Elgiloy Blue (CoCr)	0.86	1.47	1.03	0.99	1.49	0.64	1.22	1.24	1.24	0.79
BetaForce (TiMo)	1.94	1.81	2.26	2.03	2.03	1.71	1.8	1.93	2.1	1.71
Gummetal (TiNbTaZrO)	1.11	1.25	0.97	1.08	1.47	1.17	1.06	1.51	1.21	1.14

**Table 3 materials-14-06233-t003:** One-factor ANOVA results.

Df	Sum Sq	Mean Sq	F Value	Pr (>F)	Df
Group	4	8.645	2.1612	47.13	1.55 × 10^−15^
Residuals	45	2.063	0.0459	-	-

**Table 4 materials-14-06233-t004:** Levene’s test summary. Defining the homogeneity of variances.

Group Variable	Df1	Df2	F-Value	Pr (>F)
Group	45	4	2.236	0.08

**Table 5 materials-14-06233-t005:** Defining the homogeneity of variances. Results of simultaneous tests for general linear hypotheses.

Pairwise Group Comparison	Difference *	*t*-Value	Pr(>|t|)
Elgiloy Blue (CoCr)—BetaForce (TiMo)	−0.83522	−8.722	<0.0001
Gummetal (TiNbTaZrO)—BetaForce (TiMo)	−0.73392	−7.664	<0.0001
Nickel-Titanium (NiTi)—BetaForce (TiMo)	−0.78896	−8.239	<0.0001
Remanium (SS)—BetaForce (TiMo)	−1.29306	−13.503	<0.0001
Gummetal (TiNbTaZrO)—Elgiloy Blue (CoCr)	0.10130	1.058	0.826769
Nickel-Titanium (NiTi)—Elgiloy Blue (CoCr)	0.04626	0.483	0.988543
Remanium (SS)—Elgiloy Blue (CoCr)	−0.45784	−4.781	0.000196
Nickel-Titanium (NiTi)—Gummetal (TiNbTaZrO)	−0.05504	−0.575	0.978098
Remanium (SS)—Gummetal (TiNbTaZrO)	−0.55914	−5.839	<0.0001
Remanium (SS)—Nickel-Titanium (NiTi)	−0.50409	−5.264	<0.0001

*—the pairwise difference between sample group means.

**Table 6 materials-14-06233-t006:** Presence (+)/absence (−) of statistically significant differences between the sample group means (*p*-value < 0.05).

Orthodontic Wire	Remanium (SS)	Nickel-Titanium (NiTi)	Elgiloy Blue (CoCr)	BetaForce (TiMo)	Gummetal (TiNbTaZrO)
Remanium (SS)		+	+	+	+
Nickel-Titanium (NiTi)	+		−	+	−
Elgiloy Blue (CoCr)	+	−		+	−
BetaForce (TiMo)	+	+	+		+
Gummetal (TiNbTaZrO)	+	−	−	+	

**Table 7 materials-14-06233-t007:** Surface parameters for wires made from different materials.

Orthodontic Wire	RSm µm	Sq µm	Ssk -	Sz µm	Sk µm	Spk µm	Svk µm	Smr1 %	Smr2 %
Remanium	31.0	0.26	−0.72	2.28	0.56	0.20	0.39	8.5	85.0
NiTi	30.8	0.27	−0.68	3.25	0.62	0.22	0.39	7.7	86.3
Elgiloy Blue	42.7	0.89	−0.48	6.10	2.33	0.45	1.08	6.5	88.1
BetaForce	25.3	0.16	−0.41	1.75	0.36	0.16	0.22	10.5	88.9
Gummetal	33.0	0.41	0.14	3.11	1.07	0.44	0.37	10.4	91.3

## Data Availability

The data presented in this study are available on request from the corresponding author.

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
