# Peer review of "Frictional Properties of the TiNbTaZrO Orthodontic Wire—A Laboratory Comparison to Popular Archwires"

_materials, 2021, doi:10.3390/ma14216233_

Round 1

Reviewer 1 Report

The last paragraph of the Introduction does not belong in this section. Rephrasing is necessary.

The hardness of alloy wires also affects frictional resistance. Hardness values of the Ti-Nb wires are available in the literature and this could be discussed.

The authors state that “This procedure was carried out carefully to leave second-order bends out of the account.” How was this done? How could the authors exclude the third-order activations?

This limitation, as well other limitations of the present protocol (no lubrication etc), should be discussed.

It is obvious that the authors can profit from a vast language improvement for their manuscript. I suggest (preferably) either an English-speaking colleague or professional language services.

Author Response

Dear Reviewer,

Thank you very much for your feedback. The authors are grateful for your effort and valuable advices. We have tried to address all your suggestions as listed below. Please note that the “new line” numbers are referring to the line numbers in the modified manuscript.

  • The last paragraph of the Introduction does not belong in this section. Rephrasing is necessary.

As suggested, the authors revised the last paragraph of the Introduction (new lines 93-97)

  • The hardness of alloy wires also affects frictional resistance. Hardness values of the Ti-Nb wires are available in the literature and this could be discussed.

The authors appreciate this suggestion and decided to underline the correlation between hardness of wire alloys and frictional resistance in light of recent studies by writing the new paragraph in Discussion and adding proper citations to references (new lines 325-329, 494-496)

  • The authors state that “This procedure was carried out carefully to leave second-order bends out of the account.” How was this done? How could the authors exclude the third-order activations?

As recommended, the authors revised and clarified that part of Materials and Methods section (new lines 112-115)

  • This limitation, as well other limitations of the present protocol (no lubrication etc), should be discussed.

The authors revised the Discussion part by adding a new paragraph and believe that underlines more possible limitations of the present study (new lines 335-339)

  • It is obvious that the authors can profit from a vast language improvement for their manuscript. I suggest (preferably) either an English-speaking colleague or professional language services.

The authors did their best to improve the English language by revising carefully the new version of manuscript and believe they implemented all required corrections.

Kind regards

Reviewer 2 Report

The article is built well but it requires high attention in reading and evaluation.

I advice to insert in the abstract a brief reference to the statistical analysis carried out.

The introduction is complete and the materials and methods section is precise and detailed.

The results are described in particular detail.

The discussion section does not need any special changes.

The conclusions are clear.

However, I agree with the statement: "More in vitro and in vivo studies are necessary to find the best use for Gummetal archwire in orthodontic treatment.”

Author Response

Dear Reviewer,

Thank you very much for your positive feedback. The authors are grateful for your effort and valuable advice. We have tried to address all your suggestions as listed below. Please note that the “new line” numbers are referring to the line numbers in the modified manuscript.

  • I advise to insert in the abstract a brief reference to the statistical analysis carried out.

Corrected (new lines 16-20)

Kind regards